# Influence of Micro-Arc Oxidation Coatings on Stress Corrosion of AlMg6 Alloy

**DOI:** 10.3390/ma13020356

**Published:** 2020-01-12

**Authors:** Lesław Kyzioł, Aleksandr Komarov

**Affiliations:** 1Faculty of Marine Engineering, Gdynia Maritime University, 81-225 Gdynia, Poland; 2Joint Institute of Mechanical Engineering of the National Academy of Sciences of Belarus, The State Scientific Institution, 220072 Minsk, Belarus; al_kom@tut.by

**Keywords:** aluminum alloy AlMg6, Al_2_O_3_ coating, phase composition, corrosion, stress corrosion, micro-arc oxidation

## Abstract

This paper shows results of a study on the corrosion behavior of micro-arc oxidation (MAO) coatings sampled from the AlMg6 alloy. The alloy was simultaneously subjected to a corrosive environment and static tensile stress. For comparative purposes, the tests were run for both coated samples and samples without coatings. The research was conducted at a properly prepared stand; the samples were placed in a glass container filled with 3.5% NaCl aqueous solution and stretched. Two levels of tensile stress were accepted for the samples: *σ_1_* = 0.8*R*_0.2_
*σ_2_* = *R*_0.2_, and the tests were run for two time intervals: *t_1_* = 480 h and *t_2_* = 1000 h. Prolonged stress corrosion tests (lasting up to 1000 h) showed that the samples covered with ceramic coatings demonstrated significantly higher corrosion resistance than the samples without the coatings. Protective properties of the coating could be explained by its structure. Surface pores were insignificant, and their depth was very limited. The porosity level of the main coating layer was 1%. Such a structure of coating and its phase composition provided high protective properties.

## 1. Introduction

Al-Mg alloys combine good formability, rather high strength, corrosion resistance, and weldability. Therefore, such alloys are used in many structures exposed to weathering, and especially in shipbuilding and offshore structures. It should be noted that studies carried out on samples made of the AlMg6 alloy showed a good resistance to stress corrosion for this alloy, but one much lower than for alloy 5083 [1,2]. However, the strength of alloy 5083 containing 5% Mg is noticeably inferior to the strength of AlMg6 alloy.

Stress corrosion of materials manifests itself through the formation of cracks in the metal when exposed to corrosive environment and static tensile stress. The cracks appearing on the metal surface are perpendicular to the direction of tensile stress and can be of intercrystalline or mixed nature. Studies have shown that before the appearance of pronounced cracks, there is often an incubation period. The intensity of microcracks can be determined on the basis of changes in mechanical properties after time intervals of the stress corrosion test [1,2,3,4,5,6].

Studies have also shown that Al-Mg alloys with a content of Mg ≤ 3, 5% exhibit a low susceptibility to stress corrosion. This is due to the discontinuity of *β*-phase molecules at the grain boundaries, which in turn results from the low supersaturation of the solid solution [7,8,9,10].

Al-Mg alloys with a content of Mg ≥ 3, 5%, especially above 5% (AlMg5, AlMg6), with a certain state of the structure and under specific external conditions may be resistant to intercrystalline, layer, and stress corrosion [3,4,5]. The study has shown that Al−Mg alloys with a content of more than 6 wt. % of Mg can be obtained, and they show good weldability and high mechanical properties [1,9,10,11]. An increase in the Mg content results in an increase in strength, but leads to higher susceptibility to local corrosion and reduced resistance to corrosion under stress [12]. It is assumed that the grain boundaries are a favorable place for *β*-phase formation due to their low diffusion barrier (presence of defects such as dislocations and vacancies) [13]. In many cases, overexposure of Al−Mg sheets or other products to a temperature range of 70 °C to 200 °C may impede production, causing precipitation of the *β*-phase rich in Mg at the grain boundaries. These alloys are susceptible to intercrystalline, stress, or pitting corrosion, because the *β*-phase is electrochemically more active than the aluminum matrix [14,15,16,17,18].

Structures with a high Mg content resistant to corrosion are obtained through the use of complex manufacturing methods [4]. The goal of the research was to determine the impact of the protective ceramic coating on the corrosion resistance of the tested alloy. Micro-arc oxidation (MAO) is a very promising process in making tight coatings and curing of metal elements. This is a coating technique capable of forming ceramic coatings on metals such as Al, Mg, Ti, and their alloys [17,18,19,20,21,22]. This environmental friendly technique allows ceramic layers to be grown, giving the pieces high level mechanical and tribological properties and good corrosion protection in a single step of processing [23,24,25,26]. A number of studies have shown that the coatings obtained by micro-arc oxidation are an effective means of protecting aluminum alloys from corrosion [27,28,29,30,31,32,33]. The corrosion properties of the MAO coating of the BS 6082 Al alloy were tested against different immersion periods in a 0.5M NaCl solution for up to 48 h [28]. The research has shown the importance of sealing the pores in MAO coatings with the use of the sol–gel technique, enhancing the short-term corrosion resistance in 0.6M NaCl solution. The above investigation also revealed that the MAO coatings improved the corrosion resistance of the Al alloy because of the lack of defects in coatings [29].

Despite numerous studies on the protective properties of coatings, their effectiveness in protecting aluminum−magnesium alloys from stress corrosion has not been scrutinized. The major objective of the present study is to evaluate the overall effectiveness of MAO coatings in terms of resistance to aqueous stress corrosion. Regarding the above objective, MAO coatings were deposited on a AlMg6 alloy and their corrosion behavior was evaluated in 3.5% NaCl solution. In the available literature, there are only a few sources referring to the importance of protective coatings for stress corrosion [34,35]. These tests are very important because often the material is subject simultaneously to corrosive environments and stresses. In this paper the stress corrosion resistance has been evaluated using a change of mechanical properties of samples with coatings and without coatings.

## 2. Materials and Methods 

Samples of AlMg6 alloy without coating and coated with a ceramic coating by MAO were subjected to stress corrosion. The tests were carried out on identical samples without coatings and with coatings to determine the effectiveness of the ceramic coating that protects the material against stress corrosion.

Samples for the static tensile test and the stress corrosion were made from a sheet with a thickness of g = 3 mm. The samples were cut down in a direction transverse to the rolling direction. The chemical composition and parameters of heat treatment of the AlMg6 alloy plates are presented in Table 1. The shape and dimensions of the samples for the determination of mechanical and stress corrosion properties are shown in Figure 1.

The samples were polished with abrasive paper 600# and degreased with acetone followed by rinsing with distilled water before coating formation. The electrolyte was an aqueous solution of 2 g/L KOH and 5 g/L Na_2_SiO_3_. The electrolyte was agitated with compressed air. A forty litter bath from stainless steel was used. The specimen was served as the anode and the bath wall was served as the contrary electrode. The system was cooled by cold water pumped through double walls of the bath. The electrolyte temperature was controlled at 25−30 °C throughout the process. The MAO treatment was carried out using a pulsed AC power source. The current density, voltage, frequency, duty cycle, and duration time were 25A/dm^2^, 280 V, 50 Hz, 50% and 60 min, respectively. After the treatment, the samples were rinsed in distilled water and dried in air. 

Then, using abrasive paper 600#, the upper loose porous layer of about 20 μm thickness was removed from the surface of the samples, after which the samples were washed in distilled water with an ultrasonic bath SONOREX for 15 min and dried in air. The thickness of the coating on the samples prepared for testing was 150 ± 5 μm. The surface morphologies and cross-section morphologies of the coatings were examined with an optical microscope Axiovert 25 as well as with a scanning electron microscope (ZEISS, Jena, Germany). X−ray diffraction studies were carried out on a DRON−3M X−ray diffractometer (S. Petersburg, Russia) in scanning mode in 0.1 increments using Cu−Kα radiation in the Bragg–Brentano mode. The time of a set of pulses at a point was 15 s.

Stress corrosion tests of the AlMg6 alloy were carried out on samples for σ = const. The samples were placed in a glass container filled with 3.5% NaCl aqueous solution and stretched. Samples with and without coatings were subjected to stress corrosion tests for two levels of stress:

1—σ_1_ = 0.8*R*_0.2_, 2—σ_2_ = 1.0*R*_0.2_, where *R*_0.2_ is the average value of the experimentally determined yield strength of samples without coating and with coating equal to 212 MPa and 217 MPa, respectively.

The corrosion tests were conducted for two time intervals: 1—480 h, 2—1000 h. Figure 2 presents a stand for testing stress corrosion resistance.

After stress corrosion tests the samples were subjected to a static tensile test to determine the mechanical changes of the AlMg6 test material. A static tensile test on samples without coatings and with coatings was carried out on a ZwickRoell testing machine (Ulm, Germany). The research was carried out at the Faculty of Mechanical Engineering of the Gdańsk University of Technology (Gdańsk, Poland).

## 3. Research Results and Their Analysis

Figure 3 presents images of samples for mechanical and stress corrosion tests without coating and coated. The formed coating has a light gray color and a uniform surface. The study of the surface morphology of the samples showed that the surface layer of the coating is characterized by the presence of pores up to 10 microns in size (Figure 4). Such a structure is typical for coatings obtained with the use of the micro-arc oxidation method, in which there is an upper loose porous layer and a dense, low porosity basic coating layer [5,6].

Figure 5 presents images of samples without coating and coated after mechanical and stress corrosion tests. On samples without coatings, there are visible deposits and small corrosion centers (Figure 5a). There is no evidence of corrosion on samples with coating (Figure 5b). Studies performed on the cross-section of the coating after corrosion tests also showed no signs of corrosion (Figure 6).

High protective properties of the coating can be explained by its structure. As follows from the Figure 6, the pores on the surface of the coating have an insignificant size and depth and are limited to a loose surface layer. The main coating layer has a very low porosity (less than 1%), while the permeable porosity is absent. Such coating structures, as well as its phase composition, represented by chemically resistant aluminum oxide in the γ−Al_2_O_3_ and α−Al_2_O_3_ modifications (Figure 7), provide high protective properties.

On the contrary, the surface of uncoated specimens of AlMg6 alloy was subjected to intense corrosion, as evidenced by the sources of material etching and corrosion products (Figure 5a,c).

Table 2 shows results of the test for static tensile stress of the AlMg6 alloy without coatings and with coatings. The obtained results indicate identical mechanical properties of the tested materials. A very thin layer of ceramics (about 150 μm), in relation to the thickness of the sample (g_s_ = 3 mm) has no influence on the mechanical properties of the tested materials.

The stress corrosion test is very rigorous because the material is simultaneously subject to a corrosive environment and stress. None of the 24 samples tested broke. The results of the tensile test of samples after exposure to stress corrosion for two time intervals and two stress levels are presented in Table 3 and Table 4.

Table 3 shows results of the test for static tensile stress in samples with and without coatings at *t* = 480 h exposed to stress corrosion at the level of *σ* = *0.8R_0.2_* and *σ* = *R_0.2_*.

The results of the tests showed that for the level of stress *σ = 0.8R_0.2_*:-The samples without coating exhibited a decline of *R_0.2_* by 15%, *R_m_* by 6%, and *A_5_* by over 20%;-The samples with coating did not exhibit any deterioration of mechanical properties. The protective coating served its purpose well, protecting the material against corrosion.

For the level of stress *σ = R_0.2_*:-The samples without coating exhibited a decline of *R_0.2_* by 20%, *R_m_* by 8%, and *A_5_* by over 36%;-The samples with coating exhibited an insignificant decline of *R_0.2_* by 5%, no change, and *A_5_* by 7%.

For such a high level of stress the samples without coating exhibited a significant decline in mechanical properties caused by simultaneous exposure to stress and a corrosive environment.

In the case of the coated samples, there was an insignificant decline in mechanical properties caused by the high level of stress. The coating very tightly protected the sample surface, and thus the material of the sample was not subjected to corrosive environment.

Table 4 shows the results of the samples exposed to stress corrosion at similar levels of stress, i.e., *σ* = *0.8R_0.2_* and *σ* = *R_0.2_* after the time of t = 1000 h.

For the level of stress *σ* = *0.8R_0.2_*:
-the samples without coating exhibited a decline of *R_0.2_* by 25%, *R_m_* by 18%, and *A_5_* by over 50%;-the samples with coatings exhibited a decline of *R_0.2_* by 3%, *R_m_* by 6%, and *A_5_* by 14%.

With the amount of stress corrosion time, a further decline in the mechanical properties of uncoated AlMg6 alloy was observed.

The coated samples showed an insignificant decline in mechanical properties. The decline was not caused by the corrosive environment, but rather by crawling of the material, which was subjected to a high level of stress.

For the level of stress *σ* = *R_0.2_*:-The samples without coating exhibited a decline of *R_0.2_* by over 30%, *R_m_* by 23%, and *A_5_* by nearly 60%;-The samples with coatings exhibited a decline of *R_0.2_* by 4%, *R_m_* by 8%, and *A_5_* by 18%.

For such a high level of stress and a long exposure to a corrosive environment, a further decline in mechanical properties was observed in the samples with no protective coatings. The decline occurred due to a simultaneous concurrence of high stress and corrosion caused by the 3.5% NaCl solution.

Having been exposed to a corrosive environment for *t* = 1000 h and highly stressed for up to *σ* = *R_0.2_*, the coated samples sustained an insignificant decrease in mechanical properties. The limited decline is caused by the high level of stress.

Figure 8, Figure 9 and Figure 10 shows the influence of corrosive environment and stress level on the reduction of mechanical properties of AlMg6 alloy samples with and without a ceramic coating. The research shows that there was almost a 30% reduction in yield strength and tensile strength, and nearly a 60% reduction in the plasticity of AlMg6 alloy samples without a ceramic coating for stress level *σ* = *R_0.2_* and corrosion exposure time *t* = 1000 h. For identical corrosion conditions and stress levels, there was a 4% decrease in yield strength and tensile strength as well as a 8% decrease in plasticity of AlMg6 alloy samples coated with ceramic coating.

The samples of AlMg6 alloy without coatings showed a concurrence of static tensile stress and corrosive environment. This kind of corrosion−stress synergy is the very reason for the accelerated degradation of the material. However, the samples with ceramic coatings showed a high resistance to simultaneous corrosion and stress factors as the coatings protected the material against corrosion. An insignificant decrease in mechanical properties occurred due to the high level of stress. Figure 8, Figure 9 and Figure 10 shows a very narrow area of the decline in mechanical samples of coated AlMg6 alloys.

## 4. Summary

Studies have been carried out to evaluate the protective properties of the coating obtained by micro-arc oxidation on aluminum alloy AlMg6 from stress corrosion. The protective properties of the coating were evaluated by changing the mechanical characteristics (yield strength *R_0.2_*, tensile strength *R_m_* and relative elongation *A_5_*) after holding the samples without coating and coated in a 3.5% NaCl solution under a stress of *σ_1_* = *0.8R_0.2_* and *σ_2_* = *R_0.2_* for 480 h and 1000 h. The results have showed that the MAO coating provides a fairly effective protection of the aluminum alloy from stress corrosion if the selected test conditions are met. No corrosive centers were observed on samples with a ceramic coating, but small corrosion centers may be observed on samples without a coating. The effect of corrosion led to a decrease in the mechanical characteristics of the uncoated samples. So, after the 480-hour experiment at both stress levels, there was no noticeable decrease in the mechanical characteristics of coated samples, while the mechanical characteristics of uncoated samples decreased 1.1 (*R_m_*) to 1.5 (*A_5_*) times. After holding the samples for 1000 h, a more noticeable difference in mechanical characteristics was observed. In this case, the decrease in the characteristics of coated samples did not exceed 1.07 times (*R_m_*) to 1.12 times (*A_5_*), while the decrease in the properties of uncoated samples reached 1.3 (*R_m_*) to 2.4 times (*A_5_*). The high protective properties of the coating can be explained by the chemical inertness of its composition (*γ*−Al_2_O_3_ and *α*−Al_2_O_3_) and low porosity (less than 1%) in the absence of through porosity.

## Figures and Tables

**Figure 1 materials-13-00356-f001:**
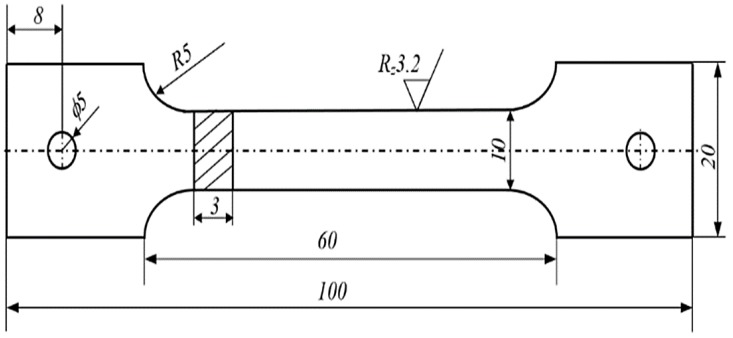
The shape and dimensions of samples for static and stress corrosion tests of the AlMg6 alloy.

**Figure 2 materials-13-00356-f002:**
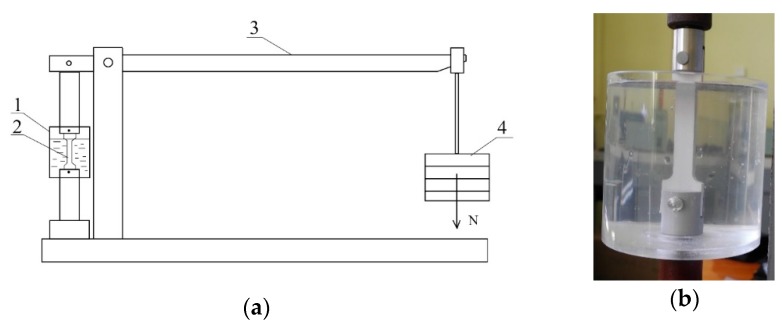
Stand for testing the stress corrosion resistance of AlMg6 alloy, (**a**) test bench scheme for σ = const., 1—container filled with 3.5% of NaCl, 2—sample, 3—lever arm, 4—load, (**b**) container filled with 3.5% of NaCl in which the sample is located.

**Figure 3 materials-13-00356-f003:**
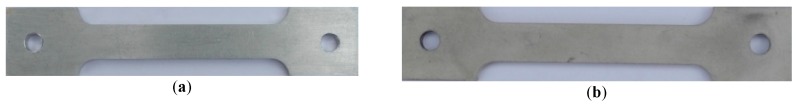
Images of samples for stress corrosion tests of AlMg6 alloy with sheet thickness g = 6 mm, (**a**) without coating, (**b**) with coating.

**Figure 4 materials-13-00356-f004:**
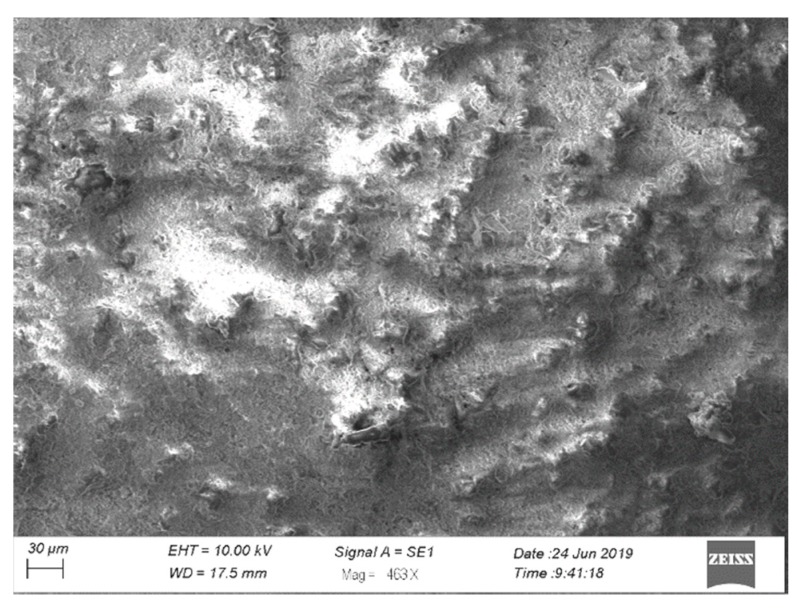
SEM image of the coating surface prior to testing.

**Figure 5 materials-13-00356-f005:**
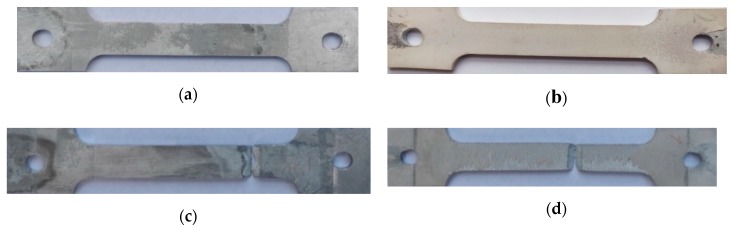
Images of samples after stress corrosion tests of AlMg6 alloy with sheet thickness g = 6 mm, (**a**) without coating, (**b**) with coating, (**c**) without cover after breaking, (**d**) with a coating after breaking.

**Figure 6 materials-13-00356-f006:**
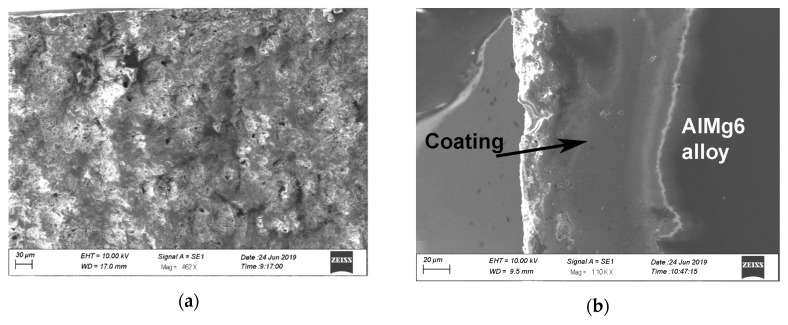
SEM image of the surface (**a**) and cross-section (**b**) of the sample with coating after stress corrosion tests.

**Figure 7 materials-13-00356-f007:**
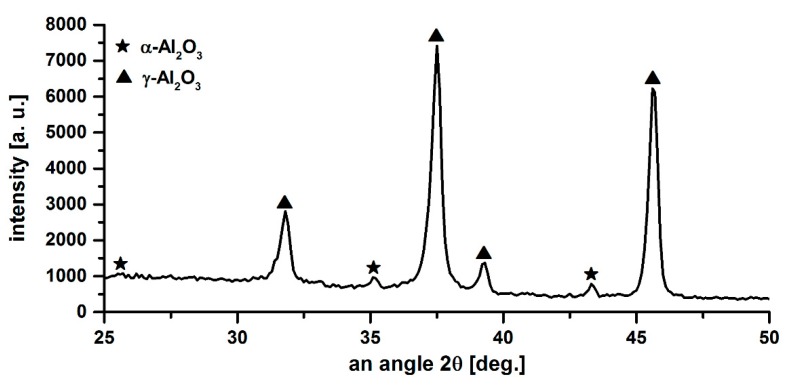
A fragment of the X−ray diffraction pattern of coating on a sample for the stress corrosion test.

**Figure 8 materials-13-00356-f008:**
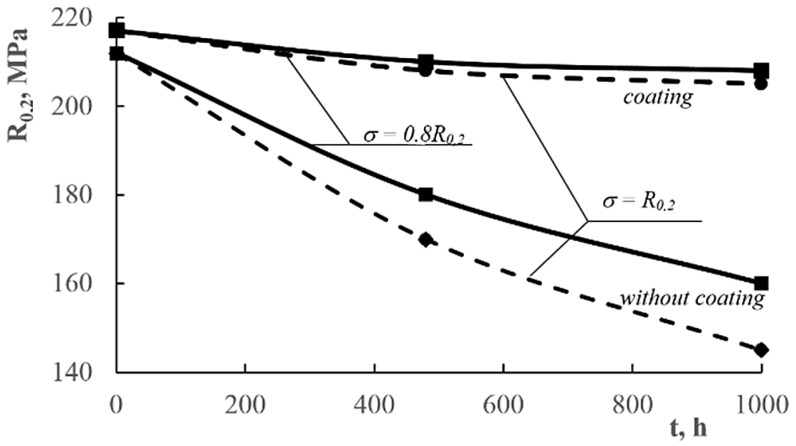
Influence of the corrosive environment and stress level on the yield strength reduction of AlMg6 alloy samples without coatings and with coatings.

**Figure 9 materials-13-00356-f009:**
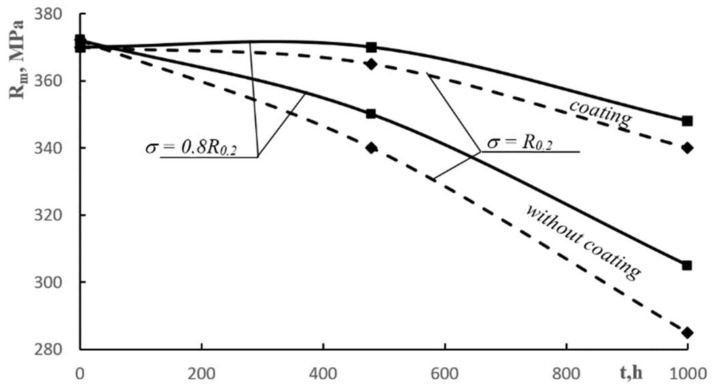
Influence of the corrosive environment and stress level on the tensile strength reduction of AlMg6 alloy samples without coatings and with coatings.

**Figure 10 materials-13-00356-f010:**
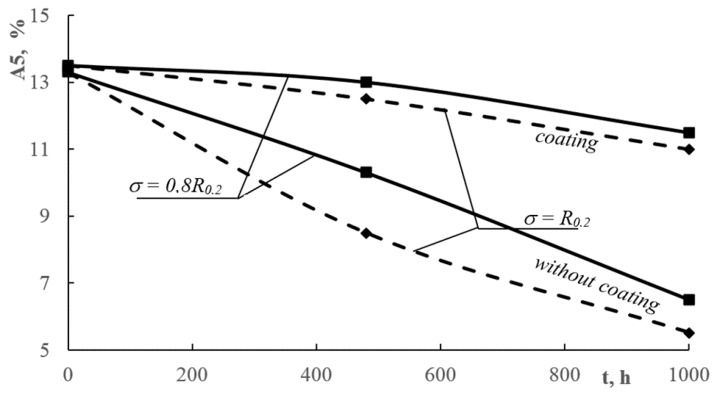
Influence of the corrosive environment and stress level on the relative elongation reduction of AlMg6 alloy samples without coatings and with coatings.

**Table 1 materials-13-00356-t001:** Chemical composition and heat treatment parameters of the AlMg6 alloy sheet.

Material	Sheet Thickness, mm	Parameters of Production Technology	Chemical Composition, %							
			Mg	Mn	Ti	Zn	Si	Fe	Cu	Al
AlMg6	6	annealing at temp. 319 °C/10 h	6.15	0.61	0.05	0.05	0.16	0.27	0.05	rest

**Table 2 materials-13-00356-t002:** Mechanical properties of AlMg6 alloy samples without coatings and with coatings.

State of the Samples.	Yield Strength *R_0.2_*	Tensile Strength *R_m_*	Relative Elongation *A_5_*
	MPa	MPa	%
without coatings	211	373	13.25
	215	372	13.45
	210	371	13.20
average	212	372	13.30
with coatings	215	369	13.65
	218	371	13.45
	218	370	13.40
average	217	370	13.50

**Table 3 materials-13-00356-t003:** Mechanical properties of samples with and without coatings of AlMg6 alloy after exposure to stress corrosion at time *t* = 480 h.

State of the Samples	Stress Level	$R_{0.2}$	$R_{m}$	$A_{5}$
		MPa	MPa	%
Without coatings	$\sigma =0.8R_{0.2}$	185	352	10.25
		182	351	10.32
		178	347	10.33
Average		180	350	10.30
With coatings		212	365	12.85
		209	374	13.18
		209	371	12.97
Average		210	370	13.00
Without coatings	$\sigma =1.0R_{0.2}$	167	338	8.45
		171	343	8.65
		172	339	8.40
Average		170	340	8.50
With coatings		208	368	12.40
		210	361	12.65
		206	366	12.45
Average		208	365	12.50

**Table 4 materials-13-00356-t004:** Mechanical properties of samples with and without coatings of AlMg6 alloy after exposure to stress corrosion at time *t* = 1000 h.

State of the Samples	Stress Level	$R_{0.2}$	$R_{m}$	$A_{5}$
		MPa	MPa	%
Without coatings	$\sigma =0.8R_{0.2}$	155	303	6.48
		162	307	6.54
		163	305	6.48
Average		160	305	6.50
With coatings		213	354	11.62
		209	342	11.48
		208	348	11.40
Average		210	348	11.50
Without coatings	$\sigma =1.0R_{0.2}$	140	288	5.48
		148	280	5.53
		147	287	5.55
average		145	285	5.52
With coatings		208	341	10.98
		205	336	11.12
		202	341	10.90
Average		205	340	11.00

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
