# Peer review of "Influence of Micro-Arc Oxidation Coatings on Stress Corrosion of AlMg6 Alloy"

_materials, 2020, doi:10.3390/ma13020356_

Round 1

Reviewer 1 Report

This paper is publishable subject to minor revisions noted. Further review is not needed.

Please clearly indicate which the thickness of the sample was used.

Line 86.

Please indicate which machining was used to obtain a 3 mm thick sample.

Line 92.

Please check the sample thickness (3 mm) indicated on Fig.1.

Line 134.

Necessary to align the thickness of the sample and the sheet.

Please rewrite this line: “Figure 3. Images of samples for stress corrosion tests of AlMg6 alloy with sheet thickness mm 6 = g.” as ”Figure 3. Images of samples for stress corrosion tests of AlMg6 alloy from 6 mm thick sheet”.

Line 144.

Please rewrite this line: “g = 6 mm, a) without coating, b) with coating, c) without cover after breaking, d) with a coating after” as “g = 6 mm, a) without coating, b) with coating, c) without coating after breaking, d) with a coating after”

Line 110.

Please indicate the mode of XRD measurement (BB or GI).

Please check to sentences: line 106 “The thickness of the coating on the samples prepared for testing was 150 ± 5 μm” and line 159: “A very thin layer of ceramics (gc = 100 ÷ 120 µm)”. Which one is correct? The literature review is mostly old. Some work in the field of MAO of Al-Mg alloys investigations is should be mentioned:

Comparison of plasma electrolytic oxidation coatings on Al alloy created in aqueous solution and molten salt electrolytes; Surf. Coat. Technol., 344 (2018) 590-595. DOI: 10.1016/j.surfcoat.2018.03.091

A universal electrolyte for the plasma electrolytic oxidation of aluminum and magnesium alloys, Materials & Design, 88 (2015) 302-309. DOI: /10.1016/j.matdes.2015.08.071

6. Please recheck the Reference List for compliance with the requirements of the Materials journal.

Author Response

Response to Reviewer 1 Comments

Point 1: Please clearly indicate which the thickness of the sample was used. 

Response 1: The thickness of the samples was 3 mm, as indicated in Figure 1. A thickness of 6 mm was indicated erroneously (now fixed).

 Point 2: Line 86. Please indicate which machining was used to obtain a 3 mm thick sample.

Response 2: Since the thickness of the samples, as specified in Response 1, was 3mm, it becomes clear from the text of the article that the samples were cut from a sheet 3 mm thick.

Point 3: Line 92. Please check the sample thickness (3 mm) indicated on Fig.1.

Response 3: As already noted, a thickness of 3 mm is correct.

Point 4: Line 134. Necessary to align the thickness of the sample and the sheet.

Please rewrite this line: “Figure 3. Images of samples for stress corrosion tests of AlMg6 alloy with sheet thickness mm 6 = g.” as ”Figure 3. Images of samples for stress corrosion tests of AlMg6 alloy from 6 mm thick sheet”.

Response 4: Thank you. The thickness value in the figure caption fixed.

Point 5: Line 144. Please rewrite this line: “g = 6 mm, a) without coating, b) with coating, c) without cover after breaking, d) with a coating after” as “g = 6 mm, a) without coating, b) with coating, c) without coating after breaking, d) with a coating after”

Response 5: The thickness value in the figure caption fixed.

Point 6: Line 110. Please indicate the mode of XRD measurement (BB or GI).

Response 6: X-ray diffraction analysis was performed in the Bragg-Brentano mode (Added in the text).

Point 7: Please check to sentences: line 106 “The thickness of the coating on the samples prepared for testing was 150 ± 5 μm” and line 159: “A very thin layer of ceramics (gc = 100 ÷ 120 µm)”. Which one is correct? The literature review is mostly old. Some work in the field of MAO of Al-Mg alloys investigations is should be mentioned:

Comparison of plasma electrolytic oxidation coatings on Al alloy created in aqueous solution and molten salt electrolytes; Surf. Coat. Technol., 344 (2018) 590-595. DOI: 10.1016/j.surfcoat.2018.03.091

A universal electrolyte for the plasma electrolytic oxidation of aluminum and magnesium alloys, Materials & Design, 88 (2015) 302-309. DOI: /10.1016/j.matdes.2015.08.071

Response 7: The coating thickness is about 150 microns. The text of the article has been updated accordingly. Two works from recent studies in the field of microarc oxidation related to the topic of this article are added to the literature review (items 33 and 34 in the References).

Point 8:  Please recheck the Reference List for compliance with the requirements of the Materials journal.

Response 8: References corrected in accordance with the requirements of the Materials journal.

 The authors thank the reviewer for comments that improved the perception of the article.

Reviewer 2 Report

This research presents a systematic study regarding the influence of MAO coating on stress corrosion of AlMg6 alloy.

The authors did a great work for example, systematic tests for stress corrosion, and analysis on change of their mechanical properties, claiming the importance of the ceramic coating on the resistance to the stress corrosion. The results clearly show the effectiveness of the coating on the stress corrosion resistance. I would accept this paper after minor revision considering below comments.    

line 41, "Al-Mg alloys with a content of Mg < 3,5 %, especially above 5% (AlMg5, AlMg6),"  Mg should be larger than 3.5%, please ckeck this.

line 115, define R0.2

Table 2, define Rm, A5

English and structure need to be improved, there are many typo and wrong formatting (some sentences need to be included in a paragraph, for example introduction)

line 228, 4. Discussion, this is not discussion, just summary.

Author Response

Response to Reviewer 2 Comments

Point 1: line 41, "Al-Mg alloys with a content of Mg < 3,5 %, especially above 5% (AlMg5, AlMg6),"  Mg should be larger than 3.5%, please check this.

Response 1: Thanks, fixed.

Point 2: line 115, define R0.2

Response 2: R0,2 is the average value of the experimentally determined yield strength of samples without coating and with coating equal to 212 MPa and 217 MPa, respectively. This explanation is included in the text of the article.

Point 3: Table 2, define Rm, A5

Response 3: Rm and A5 are defined in the Table 2.

Point 4: English and structure need to be improved, there are many typo and wrong formatting (some sentences need to be included in a paragraph, for example introduction).

Response 4: Thank you. Edits and some stylistic changes were made in the text.

Point 5: line 228, 4. Discussion, this is not discussion, just summary.

Response 5: Thanks. This is summary, of course. Fixed.

The authors thank the reviewer for comments that improved the perception of the paper.